# LEARNING A SAT SOLVER FROM SINGLE-BIT SUPERVISION

**Daniel Selsam, Matthew Lamm, Benedikt Bünz, Percy Liang, David L. Dill**
Department of Computer Science
Stanford University
Stanford, CA 94305
{dselsam,mlamm,buenz,pliang,dill}@cs.stanford.edu

**Leonardo de Moura**
Microsoft Research
Redmond, WA 98052
leonardo@microsoft.com

## ABSTRACT

We present NeuroSAT, a message passing neural network that learns to solve SAT problems after only being trained as a classifier to predict satisfiability. Although it is not competitive with state-of-the-art SAT solvers, NeuroSAT can solve problems that are substantially larger and more difficult than it ever saw during training by simply running for more iterations. Moreover, NeuroSAT generalizes to novel distributions; after training only on random SAT problems, at test time it can solve SAT problems encoding graph coloring, clique detection, dominating set, and vertex cover problems, all on a range of distributions over small random graphs.

## 1 INTRODUCTION

The propositional satisfiability problem (SAT) is one of the most fundamental problems of computer science. Cook (1971) showed that the problem is **NP**-complete, which means that searching for any kind of efficiently-checkable certificate in any context can be reduced to finding a satisfying assignment of a propositional formula. In practice, search problems arising from a wide range of domains such as hardware and software verification, test pattern generation, planning, scheduling, and combinatorics are all routinely solved by constructing an appropriate SAT problem and then calling a SAT solver (Gomes et al., 2008). Modern SAT solvers based on backtracking search are extremely well-engineered and have been able to solve problems of practical interest with millions of variables (Biere et al., 2009).

We consider the question: *can a neural network learn to solve SAT problems?* To answer, we develop a novel message passing neural network (MPNN) (Scarselli et al., 2009; Li et al., 2015; Gilmer et al., 2017), *NeuroSAT*, and train it as a classifier to predict satisfiability on a dataset of random SAT problems. We provide NeuroSAT with only a single bit of supervision for each SAT problem that indicates whether or not the problem is satisfiable. When making a prediction about a new SAT problem, we find that NeuroSAT guesses *unsatisfiable* with low confidence until it finds a solution, at which point it converges and guesses *satisfiable* with very high confidence. The solution itself can almost always be automatically decoded from the network's activations, making NeuroSAT an end-to-end SAT solver. See Figure 1 for an illustration of the train and test regimes.

Although it is not competitive with state-of-the-art SAT solvers, NeuroSAT can solve SAT problems that are substantially larger and more difficult than it ever saw during training by simply performing more iterations of message passing. Despite only running for a few dozen iterations during training, at test time NeuroSAT continues to find solutions to harder problems after hundreds and even thousands of iterations. The learning process has yielded not a traditional classifier but rather a procedure that can be run indefinitely to search for solutions to problems of varying difficulty.

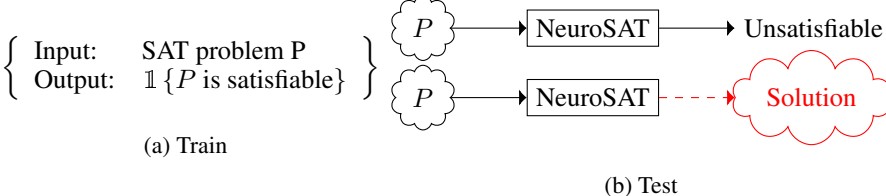

Figure 1: We train NeuroSAT to predict whether SAT problems are satisfiable, providing only a single bit of supervision for each problem. At test time, when NeuroSAT predicts *satisfiable*, we can almost always extract a satisfying assignment from the network's activations. The problems at test time can also be substantially larger, more difficult, and even from entirely different domains than the problems seen during training.

Moreover, NeuroSAT generalizes to entirely new domains. Since NeuroSAT operates on SAT problems and since SAT is **NP**-complete, NeuroSAT can be queried on SAT problems encoding any kind of search problem for which certificates can be checked in polynomial time. Although we train it using only problems from a single random problem generator, at test time it can solve SAT problems encoding graph coloring, clique detection, dominating set, and vertex cover problems, all on a range of distributions over small random graphs.

The same neural network architecture can also be used to help construct proofs for unsatisfiable problems. When we train it on a different dataset in which every unsatisfiable problem contains a small contradiction (call this trained model *NeuroUNSAT*), it learns to detect these contradictions instead of searching for satisfying assignments. Just as we can extract solutions from NeuroSAT's activations, we can extract the variables involved in the contradiction from NeuroUNSAT's activations. When the number of variables involved in the contradiction is small relative to the total number of variables, knowing which variables are involved in the contradiction can enable constructing a resolution proof more efficiently.

## 2 PROBLEM SETUP

*Background.* A formula of propositional logic is a boolean expression built using the constants true (1) and false (0), variables, negations, conjunctions, and disjunctions. A formula is *satisfiable* provided there exists an assignment of boolean values to its variables such that the formula evaluates to 1. For example, the formula $(x_1 \vee x_2 \vee x_3) \wedge \neg(x_1 \wedge x_2 \wedge x_3)$ is satisfiable because it will evaluate to 1 under every assignment that does not map $x_1$, $x_2$ and $x_3$ to the same value. For every formula, there exists an equisatisfiable formula in *conjunctive normal form* (CNF), expressed as a conjunction of disjunctions of (possibly negated) variables.[1] Each conjunct of a formula in CNF is called a *clause*, and each (possibly negated) variable within a clause is called a *literal*. The formula above is equivalent to the CNF formula $(x_1 \vee x_2 \vee x_3) \wedge (\neg x_1 \vee \neg x_2 \vee \neg x_3)$, which we can represent more concisely as $\{1|2|3, \overline{1}|\overline{2}|\overline{3}\}$. A formula in CNF has a satisfying assignment if and only if it has an assignment such that every clause has at least one literal mapped to 1. A *SAT problem* is a formula in CNF, where the goal is to determine if the formula is satisfiable, and if so, to produce a satisfying assignment of truth values to variables. We use $n$ to denote the number of of variables in a SAT problem, and $m$ to denote the number of clauses.

*Classification task.* For a SAT problem $P$, we define $\phi(P)$ to be true if and only if $P$ is satisfiable. Our first goal is to learn a classifier that approximates $\phi$. Given a distribution $\Psi$ over SAT problems, we can construct datasets $\mathcal{D}_{\text{train}}$ and $\mathcal{D}_{\text{test}}$ with examples of the form $(P, \phi(P))$ by sampling problems $P \sim \Psi$ and computing $\phi(P)$ using an existing SAT solver. At test time, we get only the problem $P$ and the goal is to predict $\phi(P)$, *i.e.* to determine if $P$ is satisfiable. Ultimately we care about the *solving task*, which also includes finding solutions to satisfiable problems.

---

[1]This transformation can be done in linear time such that the size of the resulting formula has only grown linearly with respect to the original formula (Tseitin, 1968).

## 3 MODEL

A SAT problem has a simple syntactic structure and therefore could be encoded into a vector space using standard methods such as an RNN. However, the semantics of propositional logic induce rich invariances that such a syntactic method would ignore, such as permutation invariance and negation invariance. Specifically, the satisfiability of a formula is not affected by permuting the variables (*e.g.* swapping $x_1$ and $x_2$ throughout the formula), by permuting the clauses (*e.g.* swapping the first clause with the second clause), or by permuting the literals within a clause (*e.g.* replacing the clause $1|\overline{2}$ with $\overline{2}|1$). The satisfiability of a formula is also not affected by negating every literal corresponding to a given variable (*e.g.* negating all occurrences of $x_1$ in $\{1|\overline{2}, \overline{1}|\overline{3}\}$ to yield $\{\overline{1}|\overline{2}, 1|\overline{3}\}$).

We now describe our neural network architecture, NeuroSAT, that enforces both permutation invariance and negation invariance. We encode a SAT problem as an undirected graph with one node for every literal, one node for every clause, an edge between every literal and every clause it appears in, and a different type of edge between each pair of complementary literals (*e.g.* between $x_i$ and $\overline{x_i}$). NeuroSAT iteratively refines a vector space embedding for each node by passing "messages" back and forth along the edges of the graph as described in Gilmer et al. (2017). At every time step, we have an embedding for every literal and every clause. An iteration consists of two stages. First, each clause receives messages from its neighboring literals and updates its embedding accordingly. Next, each literal receives messages from its neighboring clauses as well as from its complement, then updates its embedding accordingly. Figure 2 provides a high-level illustration of the architecture.

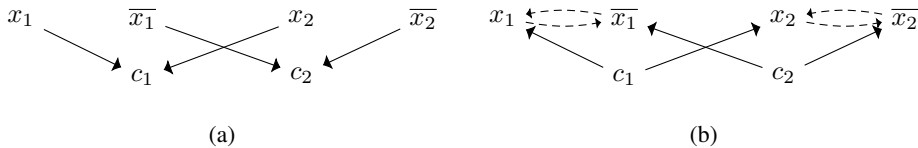

(a)  (b)

Figure 2: High-level illustration of NeuroSAT operating on the graph representation of $\{1|2, \overline{1}|\overline{2}\}$. On the top of both figures are nodes for each of the four literals, and on the bottom are nodes for each of the two clauses. At every time step $t$, we have an embedding for every literal and every clause. An iteration consists of two stages. First, each clause receives messages from its neighboring literals and updates it embedding accordingly (Figure 2a). Next, each literal receives messages from its neighboring clause as well as from its complement, and updates its embedding accordingly (Figure 2b).

More formally, our model is parameterized by two vectors ($\mathbf{L}_{\text{init}}$, $\mathbf{C}_{\text{init}}$), three multilayer perceptrons ($\mathbf{L}_{\text{msg}}$, $\mathbf{C}_{\text{msg}}$, $\mathbf{L}_{\text{vote}}$) and two layer-norm LSTMs (Ba et al., 2016; Hochreiter & Schmidhuber, 1997) ($\mathbf{L}_{\text{u}}$, $\mathbf{C}_{\text{u}}$). At every time step $t$, we have a matrix $L^{(t)} \in \mathbb{R}^{2n \times d}$ whose $i$th row contains the embedding for the literal $\ell_i$ and a matrix $C^{(t)} \in \mathbb{R}^{m \times d}$ whose $j$th row contains the embedding for the clause $c_j$, which we initialize by tiling $\mathbf{L}_{\text{init}}$ and $\mathbf{C}_{\text{init}}$ respectively. We also have hidden states $L_h^{(t)} \in \mathbb{R}^{2n \times d}$ and $C_h^{(t)} \in \mathbb{R}^{m \times d}$ for $\mathbf{L}_{\text{u}}$ and $\mathbf{C}_{\text{u}}$ respectively, both initialized to zero matrices. Let $M$ be the (bipartite) adjacency matrix defined by $M(i,j) = \mathbb{1}\{\ell_i \in c_j\}$ and let Flip be the operator that takes a matrix $L$ and swaps each row of $L$ with the row corresponding to the literal's negation. A single iteration consists of applying the following two updates:

$$(C^{(t+1)}, C_h^{(t+1)}) \leftarrow \mathbf{C}_{\text{u}}([C_h^{(t)}, M^\top \mathbf{L}_{\text{msg}}(L^{(t)})])$$

$$(L^{(t+1)}, L_h^{(t+1)}) \leftarrow \mathbf{L}_{\text{u}}([L_h^{(t)}, \text{Flip}(L^{(t)}), M\mathbf{C}_{\text{msg}}(C^{(t+1)})])$$

After $T$ iterations, we compute $L_*^{(T)} \leftarrow \mathbf{L}_{\text{vote}}(L^{(T)}) \in \mathbb{R}^{2n}$, which contains a single scalar for each literal (the literal's *vote*), and then we compute the average of the literal votes $y^{(T)} \leftarrow \text{mean}(L_*^{(T)}) \in \mathbb{R}$. We train the network to minimize the sigmoid cross-entropy loss between the logit $y^{(T)}$ and the true label $\phi(P)$.

Our architecture enforces permutation invariance by operating on nodes and edges according to the topology of the graph without any additional ordering over nodes or edges. Likewise, it enforces negation invariance by treating all literals the same no matter whether they originated as a positive or negative occurrence of a variable.

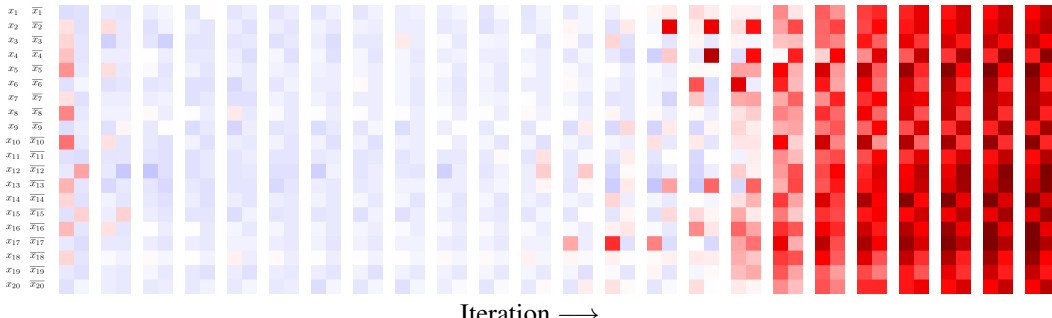

Iteration $\longrightarrow$

Figure 3: The sequence of literal votes $L_*^{(1)}$ to $L_*^{(24)}$ as NeuroSAT runs on a satisfiable problem from $\mathbf{SR}(20)$. For clarity, we reshape each $L_*^{(t)}$ to be an $\mathbb{R}^{n \times 2}$ matrix so that each literal is paired with its complement; specifically, the $i$th row contains the scalar votes for $x_i$ and $\overline{x_i}$. Here white represents zero, blue negative and red positive. For several iterations, almost every literal is voting *unsat* with low confidence (light blue). Then a few scattered literals start voting *sat* for the next few iterations, but not enough to affect the mean vote. Suddenly there is a phase transition and all the literals (and hence the network as a whole) start to vote *sat* with very high confidence (dark red). After the phase transition, the vote for each literal converges and the network stops evolving.

We stress that none of the learned parameters depend on the size of the SAT problem and that a single model can be trained and tested on problems of arbitrary and varying sizes. At both train and test time, the input to the model is simply any bipartite adjacency matrix $M$ over any number of literals and clauses. The learned parameters only determine how each individual literal and clause behaves in terms of its neighbors in the graph. Variation in problem size is handled by the aggregation operators: we sum the outgoing messages of each of a node's neighbors to form the incoming message, and we take the mean of the literal votes at the end of message passing to form the logit $y^{(T)}$.

## 4 TRAINING DATA

We want our neural network to be able to classify (and ultimately solve) SAT problems from a variety of domains that it never trained on. One can easily construct distributions over SAT problems for which it would be possible to predict satisfiability with perfect accuracy based only on crude statistics; however, a neural network trained on such a distribution would be unlikely to generalize to problems from other domains. To force our network to learn something substantive, we create a distribution $\mathbf{SR}(n)$ over pairs of random SAT problems on $n$ variables with the following property: one element of the pair is satisfiable, the other is unsatisfiable, and the two differ by negating only a single literal occurrence in a single clause. To generate a random clause on $n$ variables, $\mathbf{SR}(n)$ first samples a small integer $k$ (with mean 5) [2] then samples $k$ variables uniformly at random without replacement, and finally negates each one with independent probability 50%. It continues to generate clauses $c_i$ in this fashion, adding them to the SAT problem, and then querying a traditional SAT solver (we used Minisat Sorensson & Een (2005)), until adding the clause $c_m$ finally makes the problem unsatisfiable. Since $\{c_1, \ldots, c_{m-1}\}$ had a satisfying assignment, negating a single literal in $c_m$ must yield a satisfiable problem $\{c_1, \ldots, c_{m-1}, c'_m\}$. The pair $(\{c_1, \ldots, c_{m-1}, c_m\}, \{c_1, \ldots, c_{m-1}, c'_m\})$ are a sample from $\mathbf{SR}(n)$.

## 5 PREDICTING SATISFIABILITY

Although our ultimate goal is to solve SAT problems arising from a variety of domains, we begin by training NeuroSAT as a classifier to predict satisfiability on $\mathbf{SR}(40)$. Problems in $\mathbf{SR}(40)$ are small enough to be solved efficiently by modern SAT solvers—a fact we rely on to generate the

---

[2]We use $2 + \mathbf{Bernoulli}(0.3) + \mathbf{Geo}(0.4)$ so that we generate clauses of varying size but with only a small number of clauses of length 2, since too many random clauses of length 2 make the problems too easy on average.

problems—but the classification problem is highly non-trivial from a machine learning perspective. Each problem has 40 variables and over 200 clauses on average, and the positive and negative examples differ by negating only a single literal occurrence out of a thousand. We were unable to train an LSTM on a many-hot encoding of clauses (specialized to problems with 40 variables) to predict with >50% accuracy on its training set. Even the canonical SAT solver MiniSAT (Sorensson & Een, 2005) needs to backjump[3] almost ten times on average, and needs to perform over a hundred primitive logical inferences (*i.e.* unit propagations) to solve each problem.

We instantiated the NeuroSAT architecture described in §3 with $d = 128$ dimensions for the literal embeddings, the clause embeddings, and all the hidden units; 3 hidden layers and a linear output layer for each of the MLPs $\mathbf{L}_{\mathrm{msg}}$, $\mathbf{C}_{\mathrm{msg}}$, and $\mathbf{L}_{\mathrm{vote}}$; and rectified linear units for all non-linearities. We regularized by the $\ell_2$ norm of the parameters scaled by $10^{-10}$, and performed $T = 26$ iterations of message passing on every problem. We trained our model using the ADAM optimizer (Kingma & Ba, 2014) with a learning rate of $2 \times 10^{-5}$, clipping the gradients by global norm with clipping ratio 0.65 (Pascanu et al., 2012). We batched multiple problems together, with each batch containing up to 12,000 nodes (*i.e.* literals plus clauses). To accelerate the learning, we sampled the number of variables $n$ uniformly from between 10 and 40 during training (*i.e.* we trained on $\mathbf{SR}(\mathbf{U}(10, 40))$), though we only evaluate on $\mathbf{SR}(40)$. We trained on millions of problems.

After training, NeuroSAT is able to classify the test set correctly with 85% accuracy. In the next section, we examine how NeuroSAT manages to do so and show how we can decode solutions to satisfiable problems from its activations. Note: for the entire rest of the paper, *NeuroSAT* refers to the specific trained model that has only been trained on $\mathbf{SR}(\mathbf{U}(10, 40))$.

## 6 DECODING SATISFYING ASSIGNMENTS

Let us try to understand what NeuroSAT (trained on $\mathbf{SR}(\mathbf{U}(10, 40))$) is computing as it runs on new problems at test time. For a given run, we can compute and visualize the $2n$-dimensional vector of literal votes $L_*^{(t)} \leftarrow \mathbf{L}_{\mathrm{vote}}(L^{(t)})$ at every iteration $t$. Figure 3 illustrates the sequence of literal votes $L_*^{(1)}$ to $L_*^{(24)}$ as NeuroSAT runs on a satisfiable problem from $\mathbf{SR}(20)$. For clarity, we reshape each $L_*^{(t)}$ to be an $\mathbb{R}^{n \times 2}$ matrix so that each literal is paired with its complement; specifically, the $i$th row contains the scalar votes for $x_i$ and $\overline{x_i}$. Here white represents zero, blue negative and red positive. For several iterations, almost every literal is voting *unsat* with low confidence (light blue). Then a few scattered literals start voting *sat* for the next few iterations, but not enough to affect the mean vote. Suddenly, there is a phase transition and all the literals (and hence the network as a whole) start to vote *sat* with very high confidence (dark red). After the phase transition, the vote for each literal converges and the network stops evolving.

NeuroSAT seems to exhibit qualitatively similar behavior on every satisfiable problem that it predicts correctly. The problems for which NeuroSAT guesses *unsat* are similar except without the phase change: it continues to guess *unsat* with low-confidence for as many iterations as NeuroSAT runs for. NeuroSAT never becomes highly confident that a problem is *unsat*, and it almost never guesses *sat* on an *unsat* problem. These results suggest that NeuroSAT searches for a certificate of satisfiability, and that it only guesses *sat* once it has found one.

Let us look more carefully at the literal votes $L_*^{(24)}$ from Figure 3 after convergence. Note that most of the variables have one literal vote distinctly darker than the other. Moreover, the dark votes are all approximately equal to each other, and the light votes are all approximately equal to each other as well. Thus the votes seem to encode one bit for each variable. It turns out that these bits encode a satisfying assignment in this case, but they do not do so reliably in general. Recall from §3 that NeuroSAT projects the higher dimensional literal embeddings $L^{(T)} \in \mathbb{R}^{2n \times d}$ to the literal votes $L_*^{(T)}$ using the MLP $\mathbf{L}_{\mathrm{vote}}$. Figure 4 illustrates the two-dimensional PCA embeddings for $L^{(12)}$ to $L^{(26)}$ (skipping every other time step) as NeuroSAT runs on a satisfiable problem from $\mathbf{SR}(40)$. Blue and red dots indicate literals that are set to 0 and 1 in the satisfying assignment that it eventually finds, respectively. The blue and red dots cannot be linearly separated until the phase transition at the end, at which point they form two distinct clusters according to the satisfying assignment. We

---

[3]*i.e.* backtrack multiple steps at a time

| Trained on: | **SR**(**U**(10, 40)) |
|---:|:---|
| Trained with: | 26 iterations |
| Tested on: | **SR**(40) |
| Tested with: | 26 iterations |
| Overall test accuracy: | 85% |
| Accuracy on *unsat* problems: | 96% |
| Accuracy on *sat* problems: | 73% |
| **Percent of *sat* problems solved:** | **70%** |

Table 1: NeuroSAT's performance at test time on $\mathbf{SR}(40)$ after training on $\mathbf{SR}(\mathbf{U}(10, 40))$. It almost never guesses *sat* on unsatisfiable problems. On satisfiable problems, it correctly guesses *sat* 73% of the time, and we can decode a satisfying assignment for 70% of the satisfiable problems by clustering the literal embeddings $L^{(T)}$ as described in §6.

observe a similar clustering almost every time the network guesses *sat*. Thus the literal votes $L_*^{(T)}$ only ever encode the satisfying assignment by chance, when the projection $\mathbf{L}_{\text{vote}}$ happens to preserve this clustering.

Our analysis suggests a more reliable way to decode solutions from NeuroSAT's internal activations: 2-cluster $L^{(T)}$ to get cluster centers $\Delta_1$ and $\Delta_2$, partition the variables according to the predicate $\|x_i - \Delta_1\|^2 + \|\overline{x_i} - \Delta_2\|^2 < \|x_i - \Delta_2\|^2 + \|\overline{x_i} - \Delta_1\|^2$, and then try both candidate assignments that result from mapping the partitions to truth values. This decoding procedure (using $k$-means to find the two cluster centers) successfully decodes a satisfying assignment for over 70% of the satisfiable problems in the $\mathbf{SR}(40)$ test set. Table 1 summarizes the results when training on $\mathbf{SR}(\mathbf{U}(10, 40))$ and testing on $\mathbf{SR}(40)$.

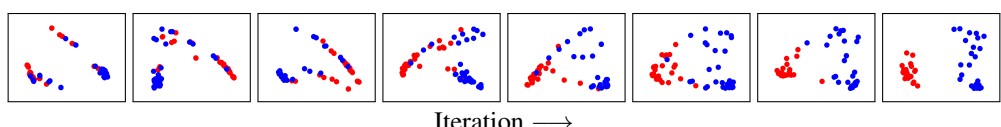

Iteration $\longrightarrow$

Figure 4: PCA projections for the high-dimensional literal embeddings $L^{(12)}$ to $L^{(26)}$ (skipping every other time step) as NeuroSAT runs on a satisfiable problem from $\mathbf{SR}(40)$. Blue and red dots indicate literals that are set to 0 and 1 in the satisfying assignment that it eventually finds, respectively. We see that the blue and red dots are mixed up and cannot be linearly separated until the phase transition at the end, at which point they form two distinct clusters according to the satisfying assignment.

Recall that at training time, NeuroSAT is only given *a single bit* of supervision for each SAT problem. Moreover, the positive and negative examples in the dataset differ only by the placement of a single edge. NeuroSAT has learned to search for satisfying assignments solely to explain that single bit of supervision.

## 7 EXTRAPOLATING TO OTHER PROBLEM DISTRIBUTIONS

### 7.1 BIGGER PROBLEMS

Even though we only train NeuroSAT on $\mathbf{SR}(\mathbf{U}(10, 40))$, it is able to solve SAT problems sampled from $\mathbf{SR}(n)$ for $n$ much larger than 40 by simply running for more iterations of message passing. Figure 5 shows NeuroSAT's success rate on $\mathbf{SR}(n)$ for a range of $n$ as a function of the number of iterations $T$. For $n = 200$, there are $2^{160}$ times more possible assignments to the variables than any problem it saw during training, and yet it can solve 25% of the satisfiable problems in $\mathbf{SR}(200)$ by running for four times more iterations than it performed during training. On the other hand, when restricted to the number of iterations it was trained with, it solves under 10% of them. Thus we see that its ability to solve bigger and harder problems depends on the fact that the dynamical system it has learned encodes generic procedural knowledge that can operate effectively over a wide range of time frames.

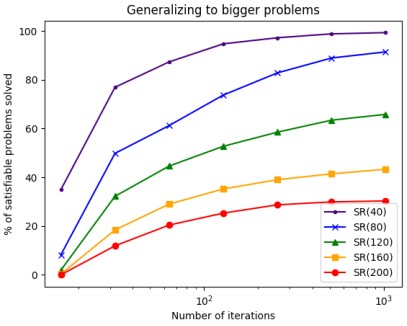

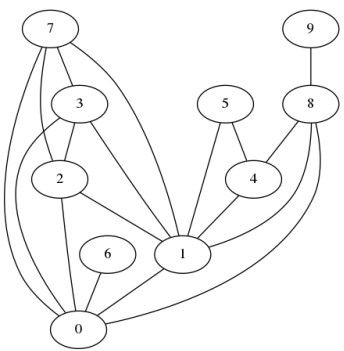

Figure 5: NeuroSAT's success rate on $\mathbf{SR}(n)$ for a range of $n$ as a function of the number of iterations $T$. Even though we only train NeuroSAT on $\mathbf{SR}(40)$ and below, it is able to solve SAT problems sampled from $\mathbf{SR}(n)$ for $n$ much larger than 40 by simply running for more iterations.

Figure 6: Example graph from the Forest-Fire distribution. The graph has a coloring for $k \geq 5$, a clique for $k \leq 3$, a dominating set for $k \geq 3$, and a vertex cover for $k \geq 6$. However, these properties are not perceptually obvious and require deliberate computation to determine.

## 7.2 DIFFERENT PROBLEMS

Every problem in **NP** can be reduced to SAT in polynomial time, and SAT problems arising from different domains may have radically different structural and statistical properties. Even though NeuroSAT has learned to search for satisfying assignments on problems from $\mathbf{SR}(n)$, we may still find that the dynamical system it has learned only works properly on problems similar to those it was trained on.

To assess NeuroSAT's ability to extrapolate to different classes of problems, we generated problems in several other domains and then encoded them all into SAT problems (using standard encodings). In particular, we started by generating one hundred graphs from each of six different random graph distributions (Barabasi, Erdös-Renyi, Forest-Fire, Random-$k$-Regular, Random-Static-Power-Law, and Random-Geometric).[4] We found parameters for the random graph generators such that each graph has ten nodes and seventeen edges on average. For each graph in each collection, we generated graph coloring problems ($3 \leq k \leq 5$), dominating-set problems ($2 \leq k \leq 4$)), clique-detection problems ($3 \leq k \leq 5$), and vertex cover problems ($4 \leq k \leq 6$).[5] We chose the range of $k$ for each problem to include the threshold for most of the graphs while avoiding trivial problems such as 2-clique. As before, we used Minisat Sorensson & Een (2005) to determine satisfiability. Figure 6 shows an example graph from the distribution. Note that the trained network does not know anything *a priori* about these tasks; the generated SAT problems need to encode not only the graphs themselves but also formal descriptions of the tasks to be solved.

Out of the 7,200 generated problems, we kept only the 4,888 satisfiable problems. On average these problems contained over two and a half times as many clauses as the problems in $\mathbf{SR}(40)$. We ran NeuroSAT for 512 iterations on each of them and found that we could successfully decode solutions for 85% of them. In contrast, Survey Propagation (SP) (Braunstein et al., 2005), the canonical (learning-free) message passing algorithm for satisfiability, does not on its own converge to a satisfying assignment on any of these problems.[6] This suggests that NeuroSAT has not simply found a way to approximate SP, but rather has synthesized a qualitatively different algorithm.

---

[4]See Newman (2010) for an overview of random graph distributions.

[5]See (Lewis, 1983) for an overview of these problems as well as the standard encodings.

[6]We implemented the version with reinforcement messages described in Knuth (2015), along with the numerical trick explained in Exercise 359.

## 8 FINDING UNSAT CORES

NeuroSAT (trained on $\mathbf{SR}(\mathbf{U}(10, 40))$) can find satisfying assignments but is not helpful in constructing proofs of unsatisfiability. When it runs on an unsatisfiable problem, it keeps searching for a satisfying assignment indefinitely and non-systematically. However, when we train the same architecture on a dataset in which each unsatisfiable problem has a small subset of clauses that are already unsatisfiable (called an *unsat core*), it learns to detect these unsat cores instead of searching for satisfying assignments. The literals involved in the unsat core can be decoded from its internal activations. When the number of literals involved in the unsat core is small relative to the total number of literals, knowing the literals involved in the unsat core can enable constructing a resolution proof more efficiently.

We generated a new distribution $\mathbf{SRC}(n, u)$ that is similar to $\mathbf{SR}(n)$ except that every unsatisfiable problem contains a small unsat core. Here $n$ is the number of variables as before, and $u$ is an unsat core over $x_1, \ldots, x_k$ ($k < n$) that can be made into a satisfiable set of clauses $u'$ by negating a single literal. We sample a pair from $\mathbf{SRC}(n, u)$ as follows. First, we initialize a problem with $u'$, and then we sample clauses (over $x_1$ to $x_n$) just as we did for $\mathbf{SR}(n)$ until the problem becomes unsatisfiable. We can now negate a literal in the final clause to get a satisfiable problem $p_s$, and then we can swap $u'$ for $u$ in $p_s$ to get $p_u$, which is unsatisfiable since it contains the unsat core $u$. We created train and test datasets from $\mathbf{SRC}(40, u)$ with $u$ sampled at random for each problem from a collection of three unsat cores ranging from three clauses to nine clauses: the unsat core $R$ from Knuth (2015), and the two unsat cores resulting from encoding the pigeonhole principles $\mathbf{PP}(2, 1)$ and $\mathbf{PP}(3, 2)$.[7] We trained our architecture on this dataset, and we refer to the trained model as *NeuroUNSAT*.

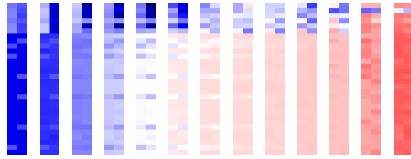 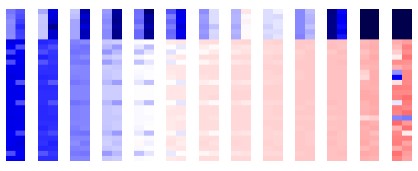

(a) NeuroUNSAT running on a satisfiable problem.

(b) NeuroUNSAT running on an unsatisfiable problem.

Figure 7: The sequence of literal votes $L_*^{(t)}$ as NeuroUNSAT runs on a pair of problems from $\mathbf{SRC}(30, \mathbf{PP}(3, 2))$. In both cases, the literals in the first six rows are involved in the unsat core. In 7a, NeuroUNSAT inspects the modified core $u'$ of the satisfiable problem but concludes that it does not match the pattern. In 7b, NeuroUNSAT finds the unsat core $u$ and votes *unsat* with high confidence (dark blue).

NeuroUNSAT is able to predict satisfiability on the test set with 100% accuracy. Upon inspection, it seems to do so by learning to recognize the unsat cores. Figure 7 shows NeuroUNSAT running on a pair of problems from $\mathbf{SRC}(30, \mathbf{PP}(3, 2))$. In both cases, the literals in the first six rows are involved in the unsat core. In Figure 7a, NeuroUNSAT inspects the modified core $u'$ of the satisfiable problem but concludes that it does not match the pattern exactly. In Figure 7b, NeuroUNSAT finds the unsat core $u$ and votes *unsat* with high confidence (dark blue). As in §6, the literals involved in the unsat core can sometimes be decoded from the literal votes $L_*^{(T)}$, but it is more reliable to 2-cluster the higher-dimensional literal embeddings $L^{(T)}$. On the test set, the small number of literals involved in the unsat core end up in their own cluster 98% of the time.

Note that we do not expect NeuroUNSAT to generalize to arbitary unsat cores: as far as we know it is simply memorizing a collection of specific subgraphs, and there is no evidence it has learned a generic procedure to prove *unsat*.

## 9 RELATED WORK

There have been many attempts over the years to apply statistical learning to various aspects of the SAT problem: restart strategies (Haim & Walsh, 2009), branching heuristics (Liang et al., 2016;

---

[7]The pigeonhole principle and the standard SAT encoding are described in Knuth (2015).

Grozea & Popescu, 2014; Flint & Blaschko, 2012), parameter tuning (Singh et al., 2009), and solver selection (Xu et al., 2008). None of these approaches use neural networks, and instead make use of both generic graph features and features extracted from the runs of SAT solvers. Moreover, these approaches are designed to assist existing solvers and do not aim to solve SAT problems on their own.

From the machine learning perspective, the closest work to ours is Palm et al. (2017), which showed that an MPNN can be trained to predict the unique solutions of Sudoku puzzles. We believe that their network's success is an instance of the phenomenon we study in this paper, namely that MPNNs can synthesize local search algorithms for constraint satisfaction problems. Evans et al. (2018) present a neural network architecture that can learn to predict whether one propositional formula entails another by randomly sampling and evaluating candidate assignments. Unlike NeuroSAT, their network does not perform heuristic search and can only work on simple problems for which random guessing is tractable. There have also been several recent papers showing that various neural network architectures can learn good heuristics for **NP**-hard combinatorial optimization problems (Vinyals et al., 2015; Bello et al., 2016; Dai et al., 2017); however, finding low-cost solutions to optimization problems requires less precise reasoning than finding satisfying assignments.

## 10 DISCUSSION

Our main motivation has been scientific: to better understand the extent to which neural networks are capable of precise, logical reasoning. Our work has definitively established that neural networks can learn to perform discrete search on their own without the help of hard-coded search procedures, even after only end-to-end training with minimal supervision. We found this result surprising and think it constitutes an important contribution to the community's evolving understanding of the capabilities and limitations of neural networks.

Although not our primary concern, we also hope that our findings eventually lead to improvements in practical SAT solving. As we stressed early on, as an end-to-end SAT solver the trained NeuroSAT system discussed in this paper is still vastly less reliable than the state-of-the-art. We concede that we see no obvious path to beating existing SAT solvers. One approach might be to continue to train NeuroSAT as an end-to-end solver on increasingly difficult problems. A second approach might be to use a system like NeuroSAT to help guide decisions within a more traditional SAT solver, though it is not clear that NeuroSAT provides any useful information before it finds a satisfying assignment. However, as we discussed in §8, when we trained our architecture on different data it learned an entirely different procedure. In a separate experiment omitted for space reasons, we also trained our architecture to predict whether there is a satisfying assignment involving each individual literal in the problem and found that it was able to predict these bits with high accuracy as well. Unlike NeuroSAT, it made both type I and type II errors, had no discernable phase transition, and could make reasonable predictions within only a few rounds. We believe that architectures descended from NeuroSAT will be able to learn very different mechanisms and heuristics depending on the data they are trained on and the details of their objective functions. We are cautiously optimistic that a descendant of NeuroSAT will one day lead to improvements to the state-of-the-art.

ACKNOWLEDGEMENTS

We thank Steve Mussmann, Alexander Ratner, Nathaniel Thomas, Vatsal Sharan and Cristina White for providing valuable feedback on early drafts. We also thank William Hamilton, Geoffrey Irving and Arun Chaganty for helpful discussions. This work was supported by Future of Life Institute grant 2017-158712.

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
