# OpenReview forum: "Learning a SAT Solver from Single-Bit Supervision"
_ICLR.cc/2019/Conference_

### Official Review · AnonReviewer3 · 2018-11-01
**review for "Learning a SAT Solver from Single-Bit Supervision"**

**Rating:** 7
**Confidence:** 3

**Review:**

This paper trains a neural network to solve the satisfiability problems. Based on the message passing neural network, it presents NeuroSAT and trains it as a classifier to predict satisfiability under a single bit of supervision. After training, NeuroSAT can solve problems that are larger and more difficult than it ever saw during training. Furthermore, the authors present a way to decode the solutions from the network's activations. Besides, for unsatisfiable problems, the paper also presents NeuroUNSAT, which learns to detect the contradictions in the form of UNSAT cores.

Relevance: this paper is likely to be of interest to a large proportion of the community for several reasons. Firstly, satisfiability problems arise from a variety of domains. This paper starts with a new angle to solve the SAT problem. Secondly, it uses neural networks in the SAT problem and establishes that neural networks can learn to perform a discrete search. Thirdly, the system used in this paper may also help improve existing SAT solvers.

Significance: I think the results are significant. For the decoding satisfying assignments section, the two-dimensional PCA embeddings are very clear. And the NeuroSAT's success rate for more significant problems and different problems has shown it's generalization ability. Finally, the sequences of literal votes in NeuroUNSAT have proved its ability to detect unsatisfied cores.

Novelty: NeuroSAT’s approach is novel. Based on message passing neural networks, it trains a neural network to learn to solve the SAT problem.

Soundness: This paper is technically sound.

Evaluation: The experimental section is comprehensive. There are a variety of graphs showing the performance and ability of your architecture. However, the theoretical analysis isn't very sufficient. For instance, why does the change of the dataset from the original SR(n) to SRC(n,u) lead to the change of the behavior of the network from searching for a satisfying assignment indefinitely to detecting the unsatisfiable cores?

Clarity: As a whole, the paper is clear. The definition of the problem, the model structure, the data generation, the training procedure, and the evaluation are all well organized. However, there is still a few points requiring more explanation. For instance, in figure 3, I am not sure whether darker value means larger value or smaller value because the authors only mentioned that white represents zero, blue negative and red positive. Also, in figure 7, I am not sure whether those black grids represent higher positive values or lower negative values.

A few questions:

What's the initialization of the two vectors the authors use for tiling operation? Does the initialization differ for different types of SAT problems?

How do the authors decide the number of iterations necessary for solving a particular SAT problem?

---

> ### Author Response · Authors · 2018-11-09
> **Response to AR3**
>
> Thank you for your comments and questions.
>
> > However, the theoretical analysis isn't very sufficient. For instance, why does the change of the dataset from the original SR(n) to SRC(n,u) lead to the change of the behavior of the network from searching for a satisfying assignment indefinitely to detecting the unsatisfiable cores?
>
> For SRC(n, u), the objective function assigns much lower cost to the parameters that detect the presence of the planted unsat cores than to the parameters that search for satisfying assignments, because unlike the latter, the former allow perfect classification of the dataset in a fixed, small number of steps. Such a simple approach is not an option on SR(n), because the cores are bigger and more diverse.
>
> > For instance, in figure 3, I am not sure whether darker value means larger value or smaller value because the authors only mentioned that white represents zero, blue negative and red positive. Also, in figure 7, I am not sure whether those black grids represent higher positive values or lower negative values.
>
> We also write in two places that "for several iterations, almost every literal is voting \emph{unsat} with low confidence (\ie light blue)".  We updated the paper to include two more similar parenthetical notes, one for "_sat_ with high confidence" and "dark red", and one for "_unsat_ with high confidence" and "dark blue". What you saw as black is just dark blue.
>
> > What's the initialization of the two vectors the authors use for tiling operation? Does the initialization differ for different types of SAT problems?
>
> It is just the parameters L_init and C_init that are learned by gradient descent at the same time as the other parameters are learned.  When a trained NeuroSAT is run on a SAT problem, no matter the size or origin, the same L_init and C_init are used.
>
> > How do the authors decide the number of iterations necessary for solving a particular SAT problem?
>
> Since the network usually converges once it finds a solution, one does not need to try to decode solutions after each round of message passing, and instead can run for a predetermined number of rounds and only check at the end. This is a desirable feature since it makes it easy to solve a very large number of SAT problems simultaneously as a single batch (e.g. on a GPU) without any problem-specific control flow. As for rules of thumb, Figure 5 provides data on how many iterations it took to solve what percentage of problems in SR(n) for a range of n. For the graph problems in S7.2, we simply ran NeuroSAT for (the somewhat arbitrary) 512 iterations on every problem.

---

### Official Review · AnonReviewer2 · 2018-11-01
**A promising approach to solve SAT problems with neural architectures**

**Rating:** 7
**Confidence:** 4

**Review:**

This paper presents the NeuroSAT architecture, which uses a deep, message passing neural net for predicting the satisfiability of CNF instances. The architecture is also able to predict a satisfiable assignment in the SAT case, and the literals involved in some minimal conflicting set of clauses (i.e. core) in the UNSAT case. The NeuroSAT architecture is based on a vector space embedding of literals and clauses, which exploits (with message passing) some important symmetries of SAT instances (permutation invariance and negation invariance). This architecture is tested on various classes of random SAT instances, involving both unstructured (RS) problems, and structured ones (e.g. graph colorings, vertex covers, dominating sets, etc.).

Overall the paper is well-motivated, and the experimental results are quite convincing. Arguably, the salient characteristic of NeuroSAT is to iteratively refine the confidence of literals voting for the SAT - or UNSAT - output, using a voting scheme on the last iteration of the literal matrix. This is very interesting, and NeuroSAT might be used to help existing solvers in choosing variable orderings for tackling hard instances, or hard queries (e.g. find a core).

On the other hand, the technical description of the architecture (sec. 3) is perhaps a little vague for having a clear intuition of how the classification task - for SAT instances - is handled in the NeuroSAT architecture. Namely, a brief description of the initial matrices (which encode the literal en clause embeddings) would be nice. Some comments on the role played by the multilayer perceptron units and the normalization units would also be welcome. The two update rules in Page 3 could be explained in more detail. For the sake of clarity, I would suggest to provide a figure for depicting a transition (from iteration t-1 to iteration t) in the architecture. As a minor comment, it would be nice (in Section 2) to define the main parameters $n$, $m$, and $d$ used in the rest of the paper.

Concerning the experimental part of the paper, Sections 4 & 5 are well-explained but, in Section 6,  the solution decoding method, inspired from PCA is a bit confusing. Specifically, we don’t know how a satisfying assignment is extracted from the last layer, and this should be explained in detail. According to Figure 4 and the comments above, it seems that a clustering method (with two centroids) is advocated, but this is not clear. In Table 1, the correlation between the accuracy on SAT instances, and the percent of SAT instances solved is not clear. Is the ratio of 70% measured on the CNF instances which have been predicted to be satisfiable? Or, is this ratio measured on the whole set of test instances? Finally, for the results established in Table 1, how many training instances and test instances have been used?

In Section 7, some important aspects related to experiments, are missing. In Sec 7.1, for SR(200) tasks, was NeuroSAT tested on the same conditions as those for SR(40) tasks? Notably, what is the input dimension $d$ of the embedding space here? (I guess that $d = 128$ is too small for such large instances). Also, how many training and test instances have been used to plot the curves in Figure 5? For the 4,888 satisfiable instances generated in Sec. 7.2, which solver have been used to determine the satisfiability of those instances (I guess it is Minisat, but this should be mentioned somewhere).

In Section 8, I found interesting the the ability of NeuroSAT in predicting the literals that participate in an UNSAT core. Indeed the problem of finding an UNSAT core in CNF instances is computationally harder than determining the satisfiability of this instance. So, NeuroSAT might be used here to help a solver in finding a core. But the notion of “confidence” should be explained in more detail in this section, and more generally, in the whole paper. Namely, it seems that in the last layer of each iteration, literals are voting for SAT (red colors) with some confidence (say $\delta$)  and voting for UNSAT (blue colors) with some confidence (say $\delta’$). Are $\delta$ and $\delta’$ correlated in the neural architecture? And, how confidences for UNSAT votes are updated?

Finally, I found that the different benchmarks where relevant, but I would also suggest (for future work, or in the appendix) to additionally perform experiments on the well-known random 3-SAT instances ($k$ is fixed to 3). Here, it is well-known that a phase transition (on the instances, not the solver/learner) occurs at 4.26 for the clause/variable ratio. A plot displaying the performance of NeuroSAT (accuracy in predicting the label of the instance) versus the clause/variable ratio would be very helpful in assessing the robustness of NeuroSAT on the so-called “hard” instances (which are close to 4.26). By extension, there have been a lot of recent work in generating “pseudo-industrial” random SAT instances, which incorporate some structure (e.g. communities) in order to mimic real-world structured SAT instances. To this point, it would be interesting to analyze the performance of NeuroSAT on such pseudo-industrial instances.

---

> ### Author Response · Authors · 2018-11-09
> **Response to AR2**
>
> Thank you for your comments and questions.
>
> > a brief description of the initial matrices (which encode the literal en clause embeddings) would be nice.
>
> The initial vectors L_init and C_init are simply parameters of the model, that we learn simultaneously with the other parameters.
>
> > For the sake of clarity, I would suggest to provide a figure for depicting a transition (from iteration t-1 to iteration t) in the architecture.
>
> We graphically depict a single iteration in Figure 2, though it is very high-level. Do you have a particular middle-ground in mind, between the figure we have and the equations themselves?
>
> > As a minor comment, it would be nice (in Section 2) to define the main parameters $n$, $m$, and $d$ used in the rest of the paper.
>
> We updated S2 to introduce n and m. We cannot introduce d there since d only makes sense in the context of the model, which is not discussed until S3.
>
> > Concerning the experimental part of the paper, Sections 4 & 5 are well-explained but, in Section 6, the solution decoding method, inspired from PCA is a bit confusing. Specifically, we don’t know how a satisfying assignment is extracted from the last layer, and this should be explained in detail. According to Figure 4 and the comments above, it seems that a clustering method (with two centroids) is advocated, but this is not clear
>
> Here is the description we give in the paper on the 2-clustering approach: "2-cluster $L^{(T)}$ to get cluster centers $\Delta_1$ and $\Delta_2$, partition the variables according to the predicate \( \| x_i - \Delta_1 \|^2 + \| \flip{x_i} - \Delta_2 \|^2 < \| x_i - \Delta_2 \|^2 + \| \flip{x_i} - \Delta_1 \|^2 \), and then try both candidate assignments that result from mapping the partitions to truth values." If you clarify what you find confusing or missing from this explanation, we will try to improve the explanation in the paper.
>
> > In Table 1, the correlation between the accuracy on SAT instances, and the percent of SAT instances solved is not clear. Is the ratio of 70% measured on the CNF instances which have been predicted to be satisfiable? Or, is this ratio measured on the whole set of test instances?
>
> As the caption says, in that experiment we were able to decode a satisfying assignment for 70% of the satisfiable problems. The satisfiable problems includes the subset of satisfiable problems for which the network incorrectly predicted _unsat_. To a first approximation, the 70% number we report means that we could decode solutions for approximately 96% of the problems correctly predicted to be _sat_; however, the 70% does include a few problems for which the network found a solution but nonetheless incorrectly guessed _unsat_. We expect this case to happen when the network finds the solution towards the very end of message passing, and does not have enough time to flip all the literal votes.
>
> Also note that the percentage of satisfiable problems solved is the metric we actually care about, whereas we only care about classification accuracy for instrumental reasons.
>
> > Finally, for the results established in Table 1, how many training instances and test instances have been used?
>
> It is easy to generate unlimited data from these distributions. We trained on millions of problems, and tested on hundreds of thousands of them.
>
> > In Sec 7.1, for SR(200) tasks, was NeuroSAT tested on the same conditions as those for SR(40) tasks? Notably, what is the input dimension $d$ of the embedding space here? (I guess that $d = 128$ is too small for such large instances).
>
> Once trained, NeuroSAT has learned parameters whose dimensions depend on the hyperparameter $d$. It is not possible to run NeuroSAT with a larger $d$ at test time. For SR(200), we use the exact same trained NeuroSAT model as in Table 1, which was trained only on SR(U(10, 40)) and has $d$ = 128.
>
> > For the 4,888 satisfiable instances generated in Sec. 7.2, which solver have been used to determine the satisfiability of those instances (I guess it is Minisat, but this should be mentioned somewhere).
>
> Yes, we used Minisat. We updated the paper to mention this, and also updated S4 to make it clear we use Minisat to generate SR(n) as well.
>
> > But the notion of “confidence” should be explained in more detail in this section, and more generally, in the whole paper.
>
> We only use the phrase "confidence" informally. The semantics of the literal votes is defined by the network architecture.
>
> > Namely, it seems that in the last layer of each iteration, literals are voting for SAT (red colors) with some confidence (say $\delta$) and voting for UNSAT (blue colors) with some confidence (say $\delta’$). Are $\delta$ and $\delta’$ correlated in the neural architecture? And, how confidences for UNSAT votes are updated?
>
> I am afraid I do not understand what you are asking. Can you please clarify your use of 'correlated' and 'updated' in the last two sentences?

---

> > ### Comment · AnonReviewer2 · 2018-11-09
> > **Some comments about the authors' response**
> >
> > > We graphically depict a single iteration in Figure 2, though it is very high-level. Do you have a particular middle- ground in mind, between the figure we have and the equations themselves?
> >
> > Yes, Figure 2 is a bit too high-level and does not provide much information about the transitions. So a bit more detailed figure, showing how equations are handled would be really helpful in understanding the neural model.
> >
> > > Here is the description we give in the paper on the 2-clustering approach: "2-cluster $L^{(T)}$ to get cluster centers $\Delta_1$ and $\Delta_2$, partition the variables according to the predicate \( \| x_i - \Delta_1 \|^2 + \| \flip{x_i} - \Delta_2 \|^2 < \| x_i - \Delta_2 \|^2 + \| \flip{x_i} - \Delta_1 \|^2 \), and then try both candidate assignments that result from mapping the partitions to truth values." If you clarify what you find confusing or missing from this explanation, we will try to improve the explanation in the paper.
> >
> > Some additional comments (maybe in the Appendix) about how this approach is implemented would be fine. Namely, how do you find the clusters? What it the algorithm here? And, finally, which assignment do you choose ( we have two candidate assignments, so which is the best)?
> >
> > > It is easy to generate unlimited data from these distributions. We trained on millions of problems, and tested on hundreds of thousands of them.
> >
> > Sure. But for the sake of clarity, it would be relevant to mention these orders of magnitude.
> >
> > >> Namely, it seems that in the last layer of each iteration, literals are voting for SAT (red colors) with some confidence (say $\delta$) and voting for UNSAT (blue colors) with some confidence (say $\delta’$). Are $\delta$ and $\delta’$ correlated in the neural architecture? And, how confidences for UNSAT votes are updated?
> > > I am afraid I do not understand what you are asking. Can you please clarify your use of 'correlated' and 'updated' in the last two sentences?
> >
> > In Sections 3-7, the learning model is focused on finding satisfying assignments.  So, all literals are voting for SAT with some confidence which is susceptible to change over transitions, and finally, an instance is predicted as UNSAT if such confidences are too small (i.e. there is no phase transition).
> >
> > Yet, according to Section 8, the framework can also be applied to core finding, which requires the literals to vote for UNSAT with high confidence (as illustrated in Figure 7). So, a natural question here is: do we have two kinds of votes (i.e. voting for SAT and trying to find a satisfying assignment, AND voting for UNSAT and trying to find a core)? If this is indeed the case, another question is to determine whether such votes are correlated: if one literal is voting for SAT with high-confidence, it will likely vote for UNSAT with low confidence.

---

> > > ### Author Response · Authors · 2018-11-17
> > > **Re: correlations**
> > >
> > > I am afraid that I still do not understand what you are asking, but I will try to address what I think might be a source of confusion.  After each round of message passing, each literal casts a _single_ scalar "vote". During training, the votes prior to round T are discarded, and then the round-T votes are averaged together and passed to the sigmoid function to estimate the probability that the problem is satisfiable. The network weights are optimized end-to-end to minimize the cross-entropy loss. When we train our architecture on SR(n), we observe empirically that these literal votes behave as we describe in S6, while when we train it on SRC(n, u), we observe empirically that the votes behave as we describe in S8. But in a given trained network, each literal still only casts a single vote at each time step.

---

> > > > ### Comment · AnonReviewer2 · 2018-11-18
> > > > **Re: Re: correlations**
> > > >
> > > > OK, thanks! This is significantly clearer and I think we’re converging. I would suggest describing this explicitly in the paper. One of the points which was misleading for me is the second paragraph of S6: “NeuroSAT never becomes highly confident that a problem is unsat, and it almost never guesses sat on an unsat problem”. In fact this only holds for the series of experiments on SR(20).

---

> > > > > ### Author Response · Authors · 2018-11-18
> > > > > **Re: Re: Re: correlations**
> > > > >
> > > > > Thank you for the suggestion. I think you may have overlooked the crucial note at the end of S5: "Note: for the entire rest of the paper, \emph{NeuroSAT} refers to the specific trained model that has only been trained on $\SR(\U(10, 40))$". We need to rely on a note like this because we use the phrase "NeuroSAT" in this way many times.  We also include an explicit reminder of this note whenever we draw attention to the role of the training data, as in the beginning on S8: "NeuroSAT (trained on $\SR(\U(10, 40))$) can find satisfying assignments but is not helpful in constructing proofs of unsatisfiability." We go on to say that "We trained our architecture on [the SRC(40, u)] dataset, and we refer to the trained model as \emph{NeuroUNSAT}." To address your concern, I have added another reminder at the beginning of S6, shortly before the sentence you quoted that you found confusing. Do you think it is sufficiently clear now? An alternative approach to preempting this confusion would be to make the dependence on the training distribution explicit in the notation, e.g. by referring to \mathrm{NeuroSAT}_{\SR(\U(10, 40))} and \mathrm{NeuroSAT}_{\SRC(40, u)}. This may leave less room for confusion, but I fear it would be rather cumbersome, especially since we only consider two different training distributions in the entire paper. What do you think?

---

### Official Review · AnonReviewer1 · 2018-11-02
**A neural architecture and analysis of predicting satisfiability with minimal supervision**

**Rating:** 7
**Confidence:** 3

**Review:**

The paper describes a general neural  network architecture for predicting satisfiability. Specifically, the contributions include an encoding for SAT problems, and predicting SAT using a message passing method, where the embeddings for literals and clauses are iteratively changed until convergence.

The paper seems significant considering that it brings together SAT solving and neural network architectures. The paper is very clearly written and quite precise about its contributions. The analysis especially figures 3,4, and 7 seems to give a nice intuitive ideas as to what the neural network is trying to do. However, one weakness is that the problems are run on a specific type of SAT problem the authors have created. Of course, the authors make it clear that the objective is not really to create a. State-of-the-art solver but rather to understand what a neural network trying to do SAT solving is capable of doing. On this front, I think the paper succeeds in doing this. One thing that was a little confusing is that why should all the literals turn to SAT (turn red) to prove SAT (as it is shown in figure 3). Is it that the neural network does not realize that it has found a SAT solution with a smaller subset of SAT literals. In other words, is it not capable of taking advantage of the problem structure.

In general though, this seemed to be an interesting paper though its practical implications are quite hard to know either in the SAT community or in the neural network community.

---

> ### Public Comment · (anonymous) · 2018-11-06
> **Likely no practical implications?**
>
> The paper (with admirable honesty) itself claims to have little to no impact on modern SAT solving. To quote, "As we stressed early on, as an end-to-end SAT solver the trained NeuroSAT system discussed in this paper is still vastly less reliable than the state-of-the-art. We concede that we see no obvious path to beating existing SAT solvers. "

---

> > ### Author Response · Authors · 2018-11-09
> > **practical implications uncertain**
> >
> > We agree with AR1 that the practical implications are quite hard to know. Following the sentences you quoted, we discuss encouraging signs, and close by saying "We are cautiously optimistic that a descendent of NeuroSAT will one day lead to improvements to the state-of-the-art."

---

> ### Author Response · Authors · 2018-11-09
> **Response to AR1**
>
> Thank you for your comments.
>
> > One thing that was a little confusing is that why should all the literals turn to SAT (turn red) to prove SAT (as it is shown in figure 3). Is it that the neural network does not realize that it has found a SAT solution with a smaller subset of SAT literals. In other words, is it not capable of taking advantage of the problem structure.
>
> Remember that the network is only trained to make the *mean* vote of the literals large on satisfiable problems and small (i.e. large and negative) on unsatisfiable problems. Thus on satisfiable problems it has a strong incentive to make all the literals vote _sat_ instead of only half of them.

---

### Public Comment · (anonymous) · 2018-10-29
**How significant is the work?**

I have two points:
1) Devlin and O’Sullivan (2008) examined the performance of a host of simple ML techniques for classifying satisfiability. Experimental results showed that Random Forest achieved very good performance (90+% accuracy for difficult large industrial SAT instances as well as for random 3-SAT and random k-SAT instances sourced from Satlib). However, the proposed deep learning based method achieves only 85% accuracy on randomly generated instances. This makes me question the significance of this work. The authors say that the data generation heuristic mentioned in the paper is for helping the neural network generalize better. I would be more convinced if the authors demonstrate the generalizability by evaluating the performance of NeuroSAT on real industrial instances. In conclusion, my main point is that after reading the work of Devlin and O’Sullivan (2008), I don't feel this work is important or significant.

2) As mentioned in the paper, for some cases, it may be possible to decode the satisfying assignments. However, this may require the graph neural network algorithm runs for many iterations. I was wondering what is the average required running time for decoding the satisfying assignments (e.g., how many seconds, ...)? Because if it takes too long time, then I would rather just use an existing off-the-shell SAT solver.

References:
[1] David Devlin and Barry O’Sullivan. B.: Satisfiability as a classification problem. In Proc. of the 19th Irish Conf. on
Artificial Intelligence and Cognitive Science, 2008.

---

> ### Public Comment · (anonymous) · 2018-11-06
> **Can the authors shed some light on this front? (Not the OP, but another interested reader)**
>
> I am curious what the authors have to say regarding this comment.

---

> ### Author Response · Authors · 2018-11-09
> **Re: How significant is the work?**
>
> First, some context. NeuroSAT solves SAT problems, it doesn't just predict satisfiability. We only report its classification accuracy in S5 to facilitate understanding, and in the rest of the paper we focus on the percent of satisfiable problems for which we can decode a solution.  Also, our main motivation has been scientific rather than to build a tool.  We wanted to better understand the extent to which neural networks are capable of precise, logical reasoning. As we state in S10, our work has definitively established that neural networks can learn to perform discrete search on their own without the help of hard-coded search procedures, even after only end-to-end training with minimal supervision.
>
> The DOS2008 paper is orthogonal to our work, but let's still consider it in detail. For a given set of SAT problems, it may be arbitrarily easy to classify satisfiability (e.g. if the _sat_ and _unsat_ problems come from different domains and have different statistical properties); however, high classification accuracy may not imply beating random on subproblems of the problems in the training set, let alone imply high accuracy on (sub)problems from other domains. Such degeneracy is an obvious concern for the "Crafted", "Industrial", and "Random" (not to be confused with "Random 3-SAT") categories in DOS2008, and the authors do not provide evidence that the classifiers trained on these categories are robust.
>
> Thus, for the rest of this comment we consider only their results in the "Random 3-SAT" category, which, although we find the wording on the bottom of page 6 to be confusing, we believe consists only of uniform random 3-SAT instances at the phase transition region that were generated using an unforced filtered method. Even for this category, for which the authors could have easily given precise semantics, they do not mention the size of the problems they used. They say that all 4,772 problems in this category are from SATLib. As of this writing, SATLib has only 3,700 uniform Random-3SAT problems in total, ranging from 20 to 250 variables, so it is not possible for us to deduce how the authors assembled their 4,772 problems or what sizes they were.
>
> There are three numbers (for each classifier) that DOS2008 provide that we will consider in more detail: the Random 3SAT "base", "all", and "+t" accuracies for the class ALL (meaning unsat and sat combined). For the "all" and "+t" categories, DOS2008 use extremely sophisticated feature extractors.  One set of features requires running two existing stochastic local search algorithms, GSAT and SAPS, multiple times each on the SAT problem. Another set of features involves solving the LP relaxation of an IP representing the SAT problem. A third set of features involves running DPLL on the SAT problem with some budget. Their feature extraction process alone took about 2 seconds on average for each of the random 3-SAT problems (aside: the feature extraction process took over an hour for one of the industrial problems). Depending on the size of the random 3-SAT problems, the solvers they ran as part of this process could have easily solved the problems within the budget and encoded their conclusions in the features themselves. Thus we cannot consider the "all" and "+t" numbers informative without more information about the sizes of the problems and the budgets for each of the feature extractors.
>
> It remains to consider the Random 3SAT "base" number, which is still extremely high for some of the classifiers (97.2% for decision trees).  For "base", they use only features 1-33, which are all syntactic properties of the SAT problem. We tried to reproduce these numbers, using sklearn to train a decision-tree classifier (default settings) and an MLP (6 50-node layers and otherwise default settings) to classify satisfiability on two different random 3-SAT distributions using exclusively these 33 features. For the first distribution, we generated 20,000 problems with 20 variables at threshold (~4.62), and for the second, we generated 10,000 problems with 50 variables at the threshold (~4.36). In each case we split the data evenly into train and test, and used Minisat to determine if the problem was satisfiable. Note that the authors only trained on fewer than 5,000 problems in total, so we are making the conditions at least as favorable for the classifier. Under these conditions, we got the following accuracies:
>
> DT, n=20, train: 100%, test: 54%
> DT, n=50, train: 100%, test: 54%
> MLP, n=20, train: 50%, test: 50%
> MLP, n=50, train: 52%, test: 51%
>
> Hyperparameter tuning might yield improvements, and of course, we could be making an error in this informal experiment. Nonetheless, especially given how remarkable the unqualified claim is of 97% test-set accuracy on hard random sat with only syntactic features, and how much crucial information is missing from DOS2008, I think the burden is on the authors of DOS2008 to clarify the experimental details and to provide reproducible code.

---

> ### Author Response · Authors · 2018-11-09
> **Re: How significant is the work? (II)**
>
> > 2) As mentioned in the paper, for some cases, it may be possible to decode the satisfying assignments. However, this may require the graph neural network algorithm runs for many iterations. I was wondering what is the average required running time for decoding the satisfying assignments (e.g., how many seconds, ...)? Because if it takes too long time, then I would rather just use an existing off-the-shell SAT solver.
>
> We explain the entire architecture in detail in the paper, as well as how many iterations we ran it for the different experiments. The number of seconds depends heavily on the hardware, which for neural networks is changing drastically every year. It also depends on how or whether the cost is amortized. As we explain in a comment above, we can solve a very large number of SAT problems simultaneously as a single batch (e.g. on a GPU) without any problem-specific control flow.
>
> As for the question of whether to use NeuroSAT or an off-the-shelf solver, we stress here as we do in the paper that the trained NeuroSAT discussed in the paper is not remotely competitive with off-the-shelf SAT solvers. We strongly recommend that you just use an existing off-the-shelf SAT solver, at least for the foreseeable future.

---

### Public Comment · ~Hector_Palacios1 · 2018-11-20
**Testing SR(40) with more than n variables**

The model described in section 3 has parameters $n$ and $m$ for the number of variables and clauses.

* What’s the maximum number of classes on the data use for SR(40)?
That would be $m$, right?
The paper says "over 200 clauses on average”, but it doesn’t say anything about the max.

* How does the model work with formulas with less than $n$ variables or $m$ clauses?
I understand the adjacency matrix $M$ would have some zero rows and columns.
What’s the impact of that during testing?

* How can you test on a formula with n > 40 or $m$ bigger than the training data of SR(40)?

---

> ### Author Response · Authors · 2018-11-24
> **Re: Testing SR(40) with more than n variables**
>
> > What’s the maximum number of classes on the data use for SR(40)?
>
> For any n >= 2, the number of clauses m in a problem from SR(n) can be arbitrarily large. To see this, note that for any given $M$, there is some probability that we sample the clause (x_1 \/ x_2) $M$ times in a row, in which case $m$ will be larger than $M$. The variance is not very large though.
>
> > How does the model work with formulas with less than $n$ variables or $m$ clauses?
> > How can you test on a formula with n > 40 or $m$ bigger than the training data of SR(40)?
>
> The parameters of the model do not depend on n or m in any way. We use d to refer to the dimensionality of the literal and clause embeddings, which is a hyperparameter that does not depend on n or m. As we explain in S3, the parameters of the NeuroSAT architecture are only the following:
>
> 1. L_init and C_init: vectors in R^d that for a given problem get duplicated 2n and m times respectively.
> 2. L_msg and C_msg:  MLPs that map R^d to R^d, and that get applied to the embeddings of each of the 2n literals and m clauses respectively.
> 3. L_update and C_update: layer-norm LSTMs whose dimensions also only depend on $d$, that get applied independently for each of the 2n literals and m clauses respectively.
> 4. L_vote: an MLP that maps R^d to R, that gets applied independently to the embeddings of each of the 2n literals.
>
> The input to the model at train and test time is any bipartite adjacency matrix $M$ over any number of literals and clauses.
>
> I added a paragraph at the end of S3 to clarify this point.

---

### Meta-Review · Area_Chair1 · 2018-12-16
**Area chair recommendation**

**Confidence:** 5
**Recommendation:** Accept (Poster)

**Metareview:**

The submission proposes a machine learning approach to directly train a prediction system for whether a boolean sentence is satisfiable.  The strengths of the paper seem to be largely in proposing an architecture for SAT problems and the analysis of the generalization performance of the resulting classifier on classes of problems not directly seen during training.

Although the resulting system cannot be claimed to be a state of the art system, and it does not have a correctness guarantee like DPLL based approaches, the paper is a nice re-introduction of SAT in a machine learning context using deep networks.  It may be nice to mention e.g. (W. Ruml. Adaptive Tree Search. PhD thesis, Harvard University, 2002) which applied reinforcement learning techniques to SAT problems.  The empirical validation on variable sized problems, etc. is a nice contribution showing interesting generalization properties of the proposed approach.

The reviewers were unanimous in their recommendation that the paper be accepted, and the review process attracted a number of additional comments showing the broader interest of the setting.